# Impact of Prenatal and Subsequent Adult Alcohol Exposure on Pro-Inflammatory Cytokine Expression in Brain Regions Necessary for Simple Recognition Memory

**DOI:** 10.3390/brainsci7100125

**Published:** 2017-09-30

**Authors:** Laurne S. Terasaki, Jaclyn M. Schwarz

**Affiliations:** Psychological and Brain Sciences, University of Delaware, 108 Wolf Hall, Newark, DE 19716, USA; laurne@uw.edu

**Keywords:** alcohol, cytokines, recognition, memory, hippocampus, prefrontal cortex, perirhinal cortex

## Abstract

Microglia, the immune cells of the brain, are important and necessary for appropriate neural development; however, activation of microglia, concomitant with increased levels of secreted immune molecules during brain development, can leave the brain susceptible to certain long-term changes in immune function associated with neurological and developmental disorders. One mechanism by which microglia can be activated is via alcohol exposure. We sought to investigate if low levels of prenatal alcohol exposure can alter the neuroimmune response to a subsequent acute dose of alcohol in adulthood. We also used the novel object location and recognition memory tasks to determine whether there are cognitive deficits associated with low prenatal alcohol exposure and subsequent adulthood alcohol exposure. We found that adult rats exposed to an acute binge-like level of alcohol, regardless of gestational alcohol exposure, have a robust increase in the expression of Interleukin (IL)-6 within the brain, and a significant decrease in the expression of IL-1β and CD11b. Rats exposed to alcohol during gestation, adulthood, or at both time points exhibited impaired cognitive performance in the cognitive tasks. These results indicate that both low-level prenatal alcohol exposure and even acute alcohol exposure in adulthood can significantly impact neuroimmune and associated cognitive function.

## 1. Introduction

Microglia are the innate immune cells of the brain and have a critical role in maintaining brain homeostasis. Microglia survey and monitor their surroundings, engage in phagocytosis of dying neural cells, and prune or refine weak synaptic connections among other neurodevelopmental processes. Their importance in the brain becomes even more evident in the event of injury, infection, or other insult. In response to these adverse conditions, microglia change their morphology from ramified, quiescent cells to amoeboid, macrophage-like active cells that release cytokines and chemokines. While activation of microglia is normal and necessary in response to an immune challenge or pathogen, prolonged or over-activation of microglia can be detrimental to neuronal function [1,2], particularly during the early periods of brain development. Inappropriate microglial activation has been linked to a number of neurodevelopmental disorders, including schizophrenia and autism spectrum disorder [3,4,5], as well as later-life cognitive disorders like Alzheimer’s disease [6,7,8,9], alcoholism and alcohol related cognitive decline [10,11,12,13].

The effects of alcohol on the body and the brain are dependent on a number of variables including sex, age, timing of consumption, and consumption pattern. Chronic alcohol consumption, like that seen in alcoholics, is generally associated with suppression of the peripheral immune system, while acute consumption is linked to an increase in immune activation and pro-inflammatory cytokine production [14]. The effects of alcohol on the neuroimmune system, specifically the cytokine response produced in the brain has not been studied in depth. Recent evidence indicates that binge levels of alcohol can activate microglia and the neuroimmune system [15,16,17,18], which may have implications for both the long-term consequences of prenatal alcohol exposure as well as the immediate consequences of alcohol consumption later in life.

Binge drinking during pregnancy is known to result in adverse effects in the developing fetus, often resulting in fetal alcohol spectrum disorder (FASD). FASD is one of the most prominent yet most preventable cause of mental retardation in the United States and affects between 2–5% of all live births [19]. Children with fetal alcohol syndrome (FAS), the most severe type of FASD, often have facial and cranial malformations and exhibit problems with learning and memory, language acquisition, and attention. While it is well-known that binge consumption of alcohol can have these detrimental effects on a fetus, there is still much debate over whether or not there is a safe amount of alcohol to consume during pregnancy. Some recent studies have suggested that low-to-moderate levels of alcohol consumption during pregnancy is not harmful to the child and their cognitive development, and could even potentially be beneficial [20,21,22,23]; however, our recent findings and others indicate that low levels of prenatal alcohol exposure can elicit cytokine production within the fetal brain and long-term changes in peripheral and neuroimmune function [7,18]. With the recent finding that alcohol activates microglia, particularly in the developing brain, we sought to determine whether low levels of alcohol exposure during early gestation in combination with acute alcohol re-exposure later, in adulthood, may precipitate cognitive deficits perhaps associated with an altered immune response in the brain.

## 2. Materials and Methods

### 2.1. Rats

Albino outbred Sprague-Dawley rats acquired from Envigo (Indianapolis, IN, USA) were used for all of the experiments described. They were housed in clear polypropylene cages on a 12:12-h light:dark cycle maintained at constant temperature and humidity, and had *ad libitum* access to food and water. All rats were kept in the University of Delaware’s Office of Laboratory Rat Medicine (OLAM) facility in accord with the Institutional Rat Care and Use Committee.

### 2.2. Breeding and Treatment

At the beginning of each experiment, female rats were bred individually with males. The presence of a sperm plug indicated day of conception, or embryonic day 1 (E1). On E10, pregnant dams were administered 2 g/kg of ethanol or the equal volume of water (0.5 mL/100 g) between 8 a.m.–9 a.m. and again four hours later, between 12 p.m.–1 p.m., each day from E10 to E16. Oral gavage was performed using flexible gavage needle (Instech, Plymouth Meeting, PA, USA, Cat. No. 1-FTP-18-75). In our experience, using these flexible needles for treatment of rats is safe, effective, and produces little stress for the rats as it requires less than 10 s of mild restraint. The blood alcohol concentrations (BACs) obtained from this dosing paradigm have been previously reported [18] and are approximately 70 mg/dL for 6 h each day. This week-long treatment period was chosen based on previous literature indicating that microglial progenitor cells migrate from the rats periphery to the central nervous system beginning around E10 [24]. Thus, the goal of this paradigm was to impact microglia during this important period of neural development. Our previous findings indicate that this pattern of alcohol exposure produces robust increases in pro-inflammatory cytokine expression within the fetal brain [18]. In order to control for litter effects, no more than one male pup and one female pup from a given litter was used in any experimental conditions described below.

### 2.3. Adulthood Ethanol Exposure

In all three experiments, adult offspring (postnatal day 90 or greater) were treated with either ethanol (4.5 g/kg) or equal volume of water (1.15 mL/100g) once between 8 a.m.–9 a.m .and again from 12 p.m. to 1 p.m. The treatment was administered using the same flexible gavage needle described above, which caused little stress or no to the rats. This dose of alcohol administration was based on extensive characterization of alcohol exposure in the rodent literature, and this dose produces BACs between 100–200 mg/dL [25,26,27].

### 2.4. Euthanasia, Perfusion, and Tissue Collection

In Experiment 1, we examined the cytokine response in the brain produced by an acute dose of alcohol administered to adult rats that had been previously treated in utero with a low dose of fetal alcohol or water (*n* = 6–9 rats/group) in order to measure the function of microglia in response to these challenges. Two hours after the last dose of alcohol or water, the adult rats were euthanized via administration of an overdose of the barbiturate Euthasol^®^ (ANADA 200-071, Penn Vet Supply, Lancaster, PA, USA) through intraperitoneal injection. The 2-h time point was selected based on previous analyses of blood alcohol concentrations indicating that peak BACs are achieved in rats within 2 h post alcohol administration [18,26,27,28]. Once under heavy anesthesia, the rats were perfused with cold 0.9% saline solution to remove blood and peripheral immune cells from the brain. Following perfusion, the whole hippocampus (HP), medial prefrontal cortex (mPFC), and perirhinal cortex were collected and these tissue samples were flash frozen on dry ice and stored at −80 °C for further processing. These brain regions were collected for analyses because they are critical in the cognitive processing of both the Novel Object Location (NOL) and the Novel Object Recognition (NOR) tasks [29,30,31,32,33]. Thus our goal was to determine whether and how alcohol exposure may influence the neuroimmune or cytokine response in these particular brain regions necessary for the cognitive tasks tested in the subsequent Experiments 2 (NOL) and 3 (NOR).

### 2.5. Quantitative Real-Time PCR (qPCR)

The expression of pro- and anti-inflammatory cytokine messenger RNA (mRNA) in the brain was analyzed using quantitative real-time PCR. Using Isol-RNA Lysis Reagent (Cat. No. FP2302700, 5 PRIME, San Fransisco, CA, USA), mRNA was extracted from brain and spleen samples. 1000 ng of mRNA samples were then subjected to DNase treatment to remove possible DNA contamination before being converted to complimentary DNA (cDNA) using QuantiTect^®^ Reverse Transcription Kit (Cat. No. 205310, Qiagen, Germantown, MD, USA).

Relative gene expression was determined using RealMasterMix^™^ Fast SYBR Kit (Cat. No. 2200830, 5 PRIME, Germantown, MD, USA) in a 10 μL reaction on a CFX96Touch^™^ real-time PCR machine (Bio-Rad, Hercules, CA, USA). The IL-6 primer was a QuantiTect^®^ Primer Assay Rn_Il6_1_SG (Cat. No. QT00182896, Qiagen, Hercules, CA, USA) obtained from Qiagen and diluted according to the protocol. Additional primers were obtained through Integrated DNA Technologies and diluted to a concentration of 13 μM. The sequence of the primers are as follows: GAPDH forward: GTTTGTGATGGGTGTGAACC, reverse: TCTTCTGAGTGGCAGTGATG (NM_017008.4); CD11b forward: CTGGGAGATGTGAATGGAG, reverse: ACTGATGCTGGCTACTGATG (NM_012711.1); IL-1β forward: GAAGTCAAGACCAAAGTGG, reverse: TGAAGTCAACTATGTCCCG (NM_031512.2); BDNF forward: ATCCCATGGGTTACACGAAGGAAG, reverse: AGTAAGGGCCCGAACATACGATTG (NM_001270638.1). GAPDH was used as a housekeeping gene since it did not differ significantly across sex or by treatment (prenatal or adult alcohol exposure). Samples were run using real-time PCR in duplicate and the average quantitative threshold amplification cycle number (C_q_) was determined from the duplicates. The 2^−ΔΔCq^ method was used to calculate relative gene expression, normalized to GAPDH.

### 2.6. Novel Object Location Task

#### Examination of NOL Memory Following Acute Alcohol Exposure during a 24 h Delay

In Experiment 2, we tested the effect of a single dose of alcohol on spatial memory in adult (P90) rats that had previously been exposed to prenatal alcohol or water using the NOL task. The paradigm is similar to what is described in [34], though with longer habituation phases and different objects and contextual cues (details provided below).

Habituation Phase. On Days 1 and 2 of the paradigm, rats underwent a 10 min habituation phase during which the rats were placed in a chamber (Stoelting Co., Wood Dale, IL, USA, 45 × 45 cm, black Plexiglas walls, and grey plastic floor) without any objects for 10 min. Both distal cues (light, door, and shapes placed on the room walls) as well as proximal cues (shapes and patterns placed on the chamber walls) were present throughout the duration of the paradigm. Exploration Phase. On Day 3, the rats were placed back in the same chamber with two identical plastic apples located diagonally across from each other for a 5-min exploration phase (Figure 1A). Immediately following the exploration phase, rats were treated with a single binge-like dose of either alcohol (4.5 g/kg) or water in equal volume (1.15 mL/100 g). Test Phase. Twenty-four hours later, in a 3-min test phase, the rats were placed in the chamber which contained one object in the same location as Day 3 and one object in a novel location. The rats were video recorded (Clover Electronics CM625 with Panasonic PLZ727 lens) throughout the experiment, and an experimenter blind to the treatment group and sex of the rat hand-scored the videos, recording the total time each rat spent exploring either object (in the new or familiar location). Exploration of each object was defined by the rat actively smelling, pawing at, or whisking with its snout directed toward the object within less than 1 cm away from the object [35]. In this Experiment 2, the groups tested were prenatally water-treated or ethanol-exposed males given water or ethanol (*n* = 12/group) on Day 3 of testing, and prenatally water-treated or ethanol-exposed females given water or ethanol (*n* = 11–12/group) on Day 3 testing.

### 2.7. Novel Object Recognition Task

#### Examination of NOR Memory Following Acute Alcohol Exposure during a 24 h Delay

In Experiment 3, we tested the effect of a single dose of alcohol on recognition memory in adult (P90) rats using the NOR task with a 24 h delay. The same cohort of rats used in Experiment 2 were also used for Experiment 3. Experiment 3 began one week after Experiment 2 had concluded. The paradigm is similar to that described above with a few differences. There were no contextual cues for the NOR task, and the objects used were yellow rubber ducks and white plastic over-the-door hooks (Figure 1B). The rats underwent the same 2-day habituation phase as in the NOL task. On Day 3 during the 5-min exploration phase, the rats were placed in a chamber that contained two identical objects: either 2 ducks or 2 hooks. Immediately following this exploration phase, the rats were, again, administered either alcohol (4.5 g/kg) or equal volume of water (1.15 mL/100 g). Twenty-four hours later, during test phase on Day 4, the rats were placed in the chamber which now contained one familiar object and one novel object. The videos obtained on the testing day were hand-scored in the same manner as in the NOL task. The groups were the same as described in the NOL task (*n* = 11–12/group).

### 2.8. Statistical Analyses

NOL and NOR Data. The percent of time that each rat spent exploring the object in a novel location (NOL) or the novel object (NOR) test was taken as a ratio of the time spent exploring both objects (discrimination ratio) as described in [36]. The data were analyzed using a one-sample *t*-test to compare discrimination ratios of each group to a ratio indicating no discrimination (0.5), an analysis that is commonly used for these particular tasks in the field see [37] for review. In order to compare differences across treatment groups, however, the data were also analyzed using a 2 × 2 × 2 ANOVA with sex, prenatal alcohol exposure, and adult alcohol exposure as factors. Significant main effects or interactions in the ANOVA (*p* < 0.05) were followed up with Tukey’s post hoc tests to determine individual group differences. All graphs represent the mean ± SEM of the discriminatory ratio for the novel object.

Real-Time qPCR Data. The relative gene expression (2^−ΔΔCq^) was analyzed across groups using a 2 × 2 × 2 ANOVA with sex, prenatal alcohol exposure, and adulthood alcohol exposure as the three factors. Following significant main effects or interactions, Tukey’s post hoc tests were run to determine individual group differences. All graphs depict the mean ± SEM of the relative gene expression for each treatment group.

## 3. Results

### 3.1. Experiment 1: Effect of Low Dose Prenatal Alcohol Exposure and Acute Adulthood Alcohol Exposure on Neuroinflammation in Male and Female Adult Offspring across Various Cognitive Brain Regions

In Experiment 1, we investigated the changes in inflammatory cytokine expression that occur within the brains of offspring previously exposed to either water or alcohol during gestation, and subsequently treated with a single dose of alcohol or water in adulthood.

In the hippocampus, we found a main effect of sex (*F*_1,50_ = 6.273; *p* = 0.016), prenatal alcohol exposure (*F*_1,50_ = 8.971; *p* = 0.005), and adulthood alcohol treatment (*F*_1,50_ = 17.292; *p* < 0.001) on the expression levels of IL-1β (Figure 2A). Specifically, females had significantly higher overall levels of IL-1β expression in the hippocampus relative to males (*p* < 0.05). Prenatal alcohol exposure resulted in a significant decrease in IL-1β expression in the hippocampus compared to water-treated controls (*p* < 0.05), while adult alcohol exposure resulted in the greatest suppression of IL-1β expression in the hippocampus compared to water-treated controls (*p* < 0.05) and prenatal alcohol exposed pups (*p* < 0.05). Analysis of IL-6 revealed a significant interaction of sex and adult alcohol treatment (*F*_1,50_ = 4.772; *p* = 0.034), in that adult alcohol exposure significantly increased IL-6 expression in the hippocampus; however, males exposed to alcohol in adulthood had significantly higher levels of IL-6 in the hippocampus compared to females given the same treatment (*p* < 0.05; Figure 2B). There were no significant effects of prenatal alcohol exposure on IL-6 expression in the hippocampus. Rats exposed to alcohol in adulthood had significantly decreased levels of CD11b, a marker of microglia, regardless of sex or prenatal alcohol exposure (*F*_1,50_ = 7.197; *p* = 0.010; Figure 2C). Although not an immune molecule, Brain Derived Neurotrophic Factor (BDNF) levels were measured for its importance in learning and memory. We found a significant interaction between sex and prenatal alcohol exposure (*F*_1,50_ = 10.594; *p* = 0.002; Figure 2D), as females exposed prenatally to alcohol had significantly higher levels of BDNF expression in the hippocampus (*p* = 0.001 relative to males exposed to alcohol prenatally), though post hoc tests revealed no other significant differences between the groups.

In the medial prefrontal cortex, we found no significant effects of any condition on IL-1β expression (Figure 3A). There was a significant three-way interaction between sex, prenatal alcohol exposure, and adulthood alcohol treatment on expression levels of IL-6 (*F*_1,62_ = 4.485; *p* = 0.039; Figure 3B). While alcohol treatment in adulthood significantly increased IL-6 expression in the medial prefrontal cortex, females exposed to alcohol during gestation and again in adulthood had higher levels of IL-6 compared to all other alcohol treated groups (*p* = 0.013). We found a significant interaction between sex and prenatal alcohol exposure on the levels of CD11b (*F*_1,62_ = 4.318; *p* = 0.042; Figure 3C). Females treated with alcohol prenatally had elevated levels of CD11b in the prefrontal cortex compared to all other groups, including males treated with alcohol prenatally (*p* < 0.05). There were no significant differences in the expression of BDNF in the medial prefrontal cortex of any treatment groups (Figure 3D).

Finally, in the perirhinal cortex, an area of the cortex important for object recognition memory, we found a significant main effect of sex as well as adult alcohol exposure on levels of IL-1β. IL-1β was significantly decreased in rats treated with alcohol in adulthood (*F*_1,47_ = 10.316; *p* = 0.003; Figure 4A). Overall, females had lower levels of IL-1β compared to males (*F*_1,47_ = 7.632; *p* = 0.009). IL-6 expression was significantly increased in rats exposed to alcohol in adulthood, independent of prenatal treatment and sex (*F*_1,47_ = 12.971; *p* = 0.001; Figure 4B). CD11b expression in the perirhinal cortex was similar to that of IL-1β in that CD11b was also significantly decreased in response to alcohol exposure in adulthood and significantly lower in females compared to males (*F*_1,47_ = 16.762; *p* < 0.001; *F*_1,47_ = 14.796; *p* < 0.001, respectively; Figure 4C). There was a significant main effect of prenatal alcohol exposure on levels of BDNF in both male and females such that exposure to alcohol during gestation increased BDNF expression in the perirhinal cortex (*F*_1,47_ = 4.812; *p* = 0.034; Figure 4D).

### 3.2. Experiment 2: Effects of Low Prenatal Alcohol Exposure and Acute Adulthood Alcohol Exposure on a Spatial Memory Task

In order to assess the impact of low prenatal alcohol exposure alone or in combination with acute alcohol exposure in adulthood on spatial recognition memory, we used the Novel Object Location (NOL) task. All rats were exposed to two identical objects for 5 min and then immediately administered either alcohol or water. Twenty-four hours later, the rats were placed back into the arena and allowed to explore an object in a familiar location and the same object in a novel location. In this task, we expect rats to investigate the novel location more if they successfully remembered the previously explored or familiar location.

We found a significant three-way interaction between sex, prenatal alcohol exposure, adulthood alcohol treatment (*F*_1,90_ = 4.224; *p* = 0.043). In addition to an ANOVA, each treatment group was analyzed individually using a *t*-test and compared to the chance value of object exploration (0.5, or no preference). We found that all rats were able to perform the NOL task at above-chance levels except for two groups: males exposed to alcohol prenatally and treated again with alcohol during the task, as well as females exposed to alcohol prenatally and administered water during the task (*p* = 0.897; *p* = 0.122, respectively; Figure 5A). These data indicate that these two groups experienced spatial memory deficits in this hippocampal-dependent task.

### 3.3. Experiment 3: Effects of Low Prenatal Alcohol Exposure and Acute Adulthood Alcohol Exposure on Recognition Memory Task

In order to test whether rats exposed prenatally to alcohol with or without subsequent alcohol exposure in adulthood had deficits in recognition memory, we used a Novel Object Recognition (NOR) task. In this task, rats explored one familiar and one novel object and we expect rats to investigate the novel object more if they successfully remembered the familiar object.

We found no main effects of sex, prenatal or adult alcohol exposure on exploration of the novel object. Using *t*-tests to compare each group’s performance to chance, we found that only males exposed to water during gestation and adulthood, as well as females treated with water in adulthood, independent of their prenatal alcohol exposure, successfully performed the task (*p* = 0.037; *p* = 0.013; *p* = 0.007, respectively; Figure 5B). None of the three other male treatment groups explored the novel object significantly more than chance. Females exposed to alcohol in adulthood, regardless of prenatal treatment, did not perform the task successfully either.

## 4. Discussion

In light of recent findings that binge alcohol exposure may activate microglia [10,11,15,16,17,18,38,39], and conflicting results that suggest no adverse effects of low-to-moderate alcohol consumption during pregnancy [20,23], we sought to determine whether prenatal alcohol exposure affects cytokine production in the brain and cognitive behaviors following subsequent adulthood alcohol exposure in adulthood. We found that a single, binge-like dose of alcohol in adulthood can produce significant changes in the expression of cytokines and neurotrophic factors in the brain. Similarly, prenatal alcohol exposure can also modulate the expression of cytokines and neurotrophic factors later in life. Most of our rats displayed robust learning in the Novel Object Location task, but this was significantly disrupted in males that had both prenatal alcohol exposure and a subsequent binge-dose of alcohol at the time of the task. In females, there was a significant effect of the prenatal alcohol exposure on this task. Using the Novel Object Recognition task, control male and female rats were capable of performing the task better than chance, but alcohol exposure in adulthood, and to some extent prenatal alcohol exposure negatively affected performance on this recognition memory task. Notably, we found minor sex differences in certain measures in the experiments performed here, but overall the effects of prenatal and adult alcohol exposure were similar.

The dose, timing of exposure, and many other factors can influence the impact that alcohol has on the body, brain and behavior. We chose to administer a low dose of alcohol during gestation that caused a maternal blood alcohol concentration of approximately 70 mg/dL (0.07 BAC) because of its physiological relevance to pregnant women [18]. Alcohol was administered to pregnant dams during embryonic days 10–16, the period during which microglial cells first infiltrate the fetal rat brain and colonize neural structures important for cognitive function including the hippocampus and prefrontal cortex [24,40].

In Experiment 1, we sought to investigate how adult offspring exposed to prenatal alcohol would respond to an acute dose of alcohol, particularly within the hippocampus, prefrontal cortex, and perirhinal cortex. These brain regions are critical for learning and memory, executive functioning, and simple object recognition memory, respectively. These brain regions, in particular, are vulnerable to small changes in homeostasis during development and even alcohol exposure neonatally and in adulthood [2,41]. We hypothesized that levels of pro-inflammatory molecules would increase in response to adulthood alcohol exposure, particularly in rats exposed to prenatal alcohol, based on previous research that showed activation of microglia following alcohol exposure [10,11,15,16,17,18,38,39]. Our results from this experiment indicate that acute alcohol exposure in adulthood had a robust and relatively rapid (within 2 h after the completion of a binge-like dose) effect on the expression of cytokines in the brain, including increased expression of interleukin-6 (IL-6) in all three brain regions, and decreased expression of IL-1β, CD11b, and brain-derived neurotrophic factor (BDNF) in the hippocampus and perirhinal cortex.

These cytokine data suggest that alcohol is acting as an inflammatory agent by causing a selective and robust increase in the expression of IL-6. Notably, the cytokine response produced following alcohol exposure is quite distinct from the classical pro-inflammatory cytokine response produced by other “typical” immune challenges, such as bacteria or endotoxins, which usually result in the increased expression of IL-1β, IL-6 and other important molecules such as Tumor Necrosis Factor (TNF) α. The function and mechanism of IL-6 in particular is complex. IL-6 can act as both a pro- and an anti-inflammatory molecule dependent upon the situation. Previous human studies have found increased levels of IL-6 circulating in alcoholics, and is thought to be one cause of pathological changes that occur in alcoholics because of its pro-inflammatory nature [42,43]. In a study by Gonzalez-Quintela et al. (2000), they found these same elevations in IL-6 in chronic alcoholics, however they did not see a change in IL-6 levels in non-alcoholics who were exposed to acute alcohol consumption [44]. This data are interesting in the context of our data which indicate that adult alcohol exposure produces a robust increase in IL-6 expression across all brain regions examined, while females exposed to alcohol during gestation had exaggerated levels of IL-6 in the prefrontal cortex if they were treated with alcohol subsequently in adulthood (Figure 3B). Thus, it is possible that early alcohol exposure may cause changes in the microglial response to subsequent alcohol consumption specifically in females. We are confident that these changes in IL-6 are the result of changes in microglial function as opposed to peripheral IL-6 measured in the brain. Prior to collecting this tissue for the expression of cytokines, we perfuse the animal with cold 0.9% saline, which removes peripheral immune cells from the brain. Previous papers have shown that when neural tissue is perfused of peripheral immune cells, the resultant immune cell population is almost 96% microglia, with just a small percentage of perivascular macrophages [45]. Thus we are confident that the changes in gene expression that we are measuring are the result of changes in microglial function. Interestingly, IL-1β and CD11b expression levels were decreased in rats treated with binge-like levels of alcohol in adulthood, an effect that also indicates robust changes in microglia function following acute alcohol exposure. These data do not indicate, per se, that microglia are inhibited by alcohol, but rather that the neuroimmune response to alcohol may be “alternate” or M2, which has been previously shown by others [15,17,46]. Typically, it has been thought that “M2 cytokine responses” are important for repair and restoration of the tissue [47]. Taken together, these results highlight how varied the microglial or neuroimmune response can be. This response is likely dependent upon the “immunogenic” challenge that occurs, and the subsequent role of this immune response in attempting to restore homeostasis within the brain. In future experiments, the collection of additional time-points following binge-like alcohol administration in adulthood may reveal the full time course of these changes that occur in cytokine expression within the brain following acute alcohol exposure. Such findings would help to understand the cellular and neural mechanisms that underlie cognitive disorders associated with alcohol abuse and lead to addiction.

In Experiment 2, we hypothesized that if adult alcohol exposure produces a neuroinflammatory response, it may interfere with the memory consolidation process necessary for a simple recognition memory test, such as the novel object location, particularly if rats were exposed to prenatal alcohol. Our results showed that all treatment groups, except for two, explored the object in a novel location significantly more than chance. Specifically, males who were exposed to alcohol both during gestation and again in adulthood had difficulty performing this cognitive task. Similarly, females exposed to alcohol during gestation and water during adulthood did not explore the novel location significantly more than chance. These data support our overall hypothesis that alcohol exposure (even at low prenatal doses) can interfere with simple cognitive tasks later in life. Moreover, these data highlight that males and female differ in their response to prenatal and adulthood alcohol exposure and this results in differences on this particular hippocampal-dependent task. Contrary to our expectations, a second binge-like dose of alcohol had adverse effects on males’ cognitive performance, while the same treatment group in females showed no problems in remembering the familiarly-located object. In fact, they performed better than the females that had only been exposed to alcohol prenatally. The NOL task, in particular, is highly dependent upon the hippocampus and prefrontal cortex [32,33,34]. One might hypothesize that the changes in cytokine expression that we observed in these two brain regions in Experiment 1 may explain some of these findings. In particular, if IL-1β is essential to proper learning and memory, and its levels are inhibited by both prenatal and adult alcohol exposure, this may increase the likelihood of cognitive deficits in the NOL task. Interestingly, the overall levels of IL-1β are still much higher in females than in males (Figure 2A), thus making females perhaps less vulnerable to the cognitive effects of this particular cytokine, particularly dependent upon the levels of other cytokines that are produced (e.g., IL-6). In particular, one might hypothesize that IL-6 may enhance learning and memory in this particular task, but only when increased to a certain level; importantly the levels produced by the “two hits” in female rats resulted in significantly enhanced levels of IL-6 in the prefrontal cortex (Figure 3B), thus potentially enhancing cognitive performance in female rats exposed to alcohol both prenatally and in adulthood. Future experiments that seek to inhibit microglial activation or the production of specific cytokines, such as IL-6, may better inform the specific function of microglia and the molecules they produce in the associated cognitive deficits measured here. In conclusion, the results from the NOL tests indicate that even low amounts of alcohol exposure during gestation can have lasting cognitive effects, with or without subsequent alcohol exposure.

To investigate whether there were changes in memory function for a non-hippocampal-dependent task, we used a novel object recognition task in Experiment 3. We predicted that similar to Experiment 2, in the event of neuroinflammation from prenatal and/or adulthood alcohol exposure, rats would have impaired memory consolidation and thus exhibit less exploration of the novel object. Our results show that only males exposed to water during gestation and adulthood and females exposed to water in adulthood, regardless of prenatal treatment, explored the novel object at a rate significantly greater than chance. Interestingly, rats exposed to alcohol both prenatally and in adulthood showed the least amount of exploration, indicating that alcohol exposure at both time points can significantly alter the function of memory-related brain regions. These results, in combination with others’ work in humans and rats [48,49,50] indicate that even low amounts of maternal alcohol consumption can be harmful to cognitive processes despite not eliciting fetal alcohol syndrome. One caveat of the NOR studies is that the same rats were used in both Experiment 2 (NOL) and Experiment 3 (NOR). Thus, it may be possible that the previous behavioral experience in Experiment 2 interfered with the recognition memory in Experiment 3. The overall exploration ratios in the NOR task were quite low, which is evidence that might support this caveat. That said, there were certain groups of rats, including the control males and females, that were able to perform the NOR task above chance, suggesting that the deficits seen in the alcohol exposed groups were not an artifact of interference due to performing two similar cognitive tasks, but rather an effect specific to the alcohol exposures, which produced robust inhibition in IL-1β and robust increases in IL-6 in the perirhinal cortex. We hope future experiments will uncover why the NOR task may be more vulnerable to the effects of the immune activation or alcohol exposure compared to the hippocampal-dependent NOL task.

## 5. Conclusions

Our research delves into the contentious topic of alcohol consumption during pregnancy and the potential neuroimmunological effects that it may have alone, as well as in combination with alcohol exposure later in adulthood. Our experiments characterized the expression of important pro-inflammatory and neurotrophic molecules, as well as cognitive ability in these offspring following prenatal and adult alcohol exposure. The results suggest that both males and females had an increase in cytokine production in the brain as a result of alcohol exposure in adulthood, as indicated by an increase in the expression of IL-6 across multiple cognitive brain regions. Additionally, learning and memory deficits were seen in males and females when exposed to alcohol, either during gestation, adulthood, or both time points. While further research should be conducted, these results suggest that even low amounts of alcohol during pregnancy can have lasting impacts on offspring neuroimmune function and cognition, and that alcohol consumption during adulthood can have its own deleterious effects.

## Figures and Tables

**Figure 1 brainsci-07-00125-f001:**
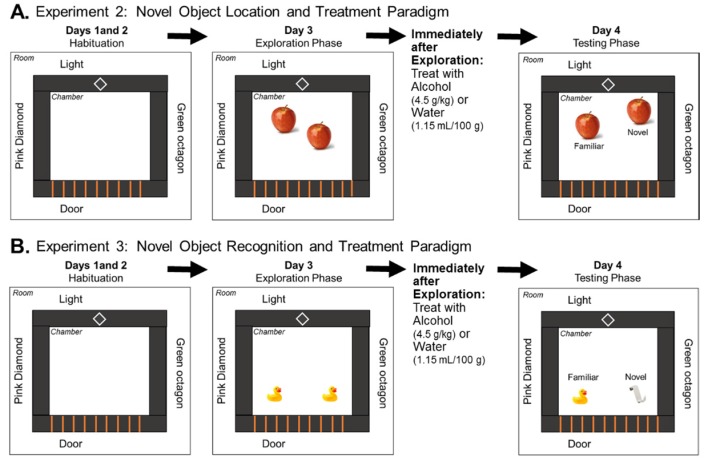
Experimental Paradigm for Novel Object Location (Experiment 2) and Novel Object Recognition (Experiment 3). Male and female rats were treated prenatally with alcohol or water and allowed to grow up undisturbed until adulthood. (**A**) Novel Object Location began with two days of habituation, followed by an exploration phase on Day 3 wherein the rats were exposed to two identical objects. Immediately after the exploration phase, the rats were treated with either alcohol (4.5 g/kg) or water (1.15 mL/100 g). Twenty four hours later, on Day 4, the rats were placed back in the chamber, however, in this testing phase one of the objects is moved to a new location. (**B**) Novel Object Recognition began with two days of habituation, followed by an exploration phase on Day 3 wherein the rats were exposed to two identical objects. Immediately after the exploration phase, the rats were treated with either alcohol (4.5 g/kg) or water (1.15 mL/100 g). Twenty-four hours later, on Day 4, the rats were placed back in the chamber, however, in this testing phase one of the objects is replaced by a novel object.

**Figure 2 brainsci-07-00125-f002:**
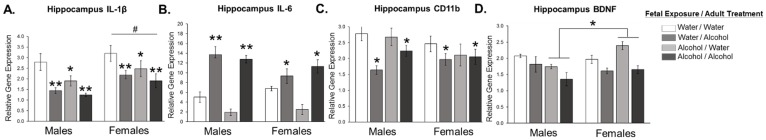
Hippocampal gene expression in male and female rats exposed to alcohol during fetal development, adulthood, or both. Pregnant dams were treated with alcohol or water on E10-16 (2 g/kg, 2×/day). Offspring were allowed to grow up undisturbed at which point they were treated with either an acute dose of alcohol (4.5 g/kg, 2×) or water. Brain tissue was collected 2 h after the last dose of alcohol for the analysis of various cytokines and Brain Derived Neurotrophic Factor (BDNF). (**A**) Analysis of IL-1β mRNA revealed a significant main effect of sex (*F*_1,50_ = 6.273; ^#^
*p* = 0.016), prenatal alcohol exposure (*F*_1,50_ = 8.971; *p* = 0.005), and adulthood alcohol treatment (F_1,50_ = 17.292; *p* < 0.001). The * indicate the main effect of prenatal alcohol exposure and the ** indicates *p* < 0.05, revealing the additive effect of both the prenatal and adult alcohol exposures relative to water treated controls and alcohol/water treated rats. (**B**) Analysis of IL-6 mRNA revealed a significant interaction of sex and adult alcohol treatment (*F*_1,50_ = 4.772; *p* = 0.034) * *p* < 0.05 indicating these males had significantly more IL-6 expression relative to untreated and alcohol/water treated rats. ** *p* < 0.05 indicating that these alcohol treated females had significantly higher levels of IL-6 relative to all other groups. (**C**) Analysis of CD11b mRNA revealed a significant main effect of adult alcohol treatment (*F*_1,50_ = 7.197; * *p* = 0.010 relative to adult water treated rats).(**D**) Analysis of BDNF mRNA in the hippocampus revealed a significant interaction between sex and prenatal alcohol exposure (*F*_1,50_ = 10.594; *p* = 0.002), * *p* < 0.05 compared to male rats exposed to alcohol prenatally.

**Figure 3 brainsci-07-00125-f003:**
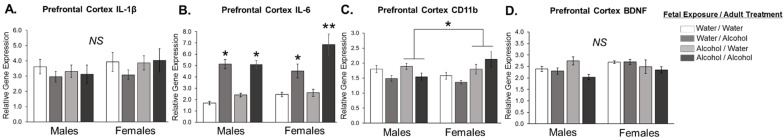
Medial Prefrontal Cortex gene expression in male and female rats exposed to alcohol during fetal development, adulthood, or both. Pregnant dams were treated with alcohol or water on E10-16 (2 g/kg, 2×/day). Offspring were allowed to grow up undisturbed at which point they were treated with either an acute dose of alcohol (4.5 g/kg, 2×) or water. Brain tissue was collected 2 h after the last dose of alcohol for the analysis of various cytokines and Brain Derived Neurotrophic Factor (BDNF). (**A**) Analysis of IL-1β mRNA revealed no significant differences (NS). (**B**) Analysis of IL-6 mRNA revealed a significant three-way interaction between sex, prenatal alcohol exposure, and adulthood alcohol treatment on expression levels of IL-6 (*F*_1,62_ = 4.485; *p* = 0.039) and post hoc tests revealed that adult alcohol treatment significantly increased IL-6 expression relative to water treated controls (* *p* < 0.05), however, females exposed to alcohol prenatally and then again in adulthood had significantly higher levels of IL-6 than all other treatment groups (** *p* < 0.05). (**C**) Analysis of CD11b mRNA revealed a significant interaction between sex and prenatal alcohol exposure (*F*_1,62_ = 4.318; *p* = 0.042), * *p* < 0.05 compared to male rats exposed to alcohol prenatally. (**D**) Analysis of BDNF mRNA in the mPFC revealed no significant differences (NS).

**Figure 4 brainsci-07-00125-f004:**
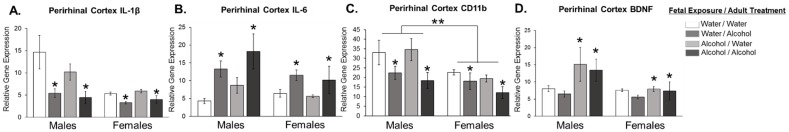
Perirhinal gene expression in male and female rats exposed to alcohol during fetal development, adulthood, or both. Pregnant dams were treated with alcohol or water on E10-17 (2 g/kg, 2×/day). Offspring were allowed to grow up undisturbed at which point they were treated with either an acute dose of alcohol (4.5 g/kg, 2×) or water. Brain tissue was collected 2 h after the last dose of alcohol for the analysis of various cytokines and Brain Derived Neurotrophic Factor (BDNF). (**A**) Analysis of IL-1β mRNA revealed a significant main effect of adult alcohol treatment (*F*_1,47_ = 10.316; * *p* = 0.003 compared to adult water treated controls) (**B**) Analysis of IL-6 mRNA revealed a significant main effect of adult alcohol treatment (*F*_1,47_ = 12.971;* *p* = 0.001 compared to adult water treated rats) (**C**) Analysis of CD11b mRNA revealed a main effect of adult alcohol treatment (*F*_1,47_ = 16.762; * *p* < 0.001 compared to adult water treated rats) and sex (*F*_1,47_ = 14.796; ** *p* < 0.001 in males compared to females).(**D**) Analysis of BDNF mRNA in the perirhinal cortex revealed a significant main effect of prenatal alcohol exposure (*F*_1,47_ = 4.812; * *p* = 0.034 in rats exposed to alcohol prenatally compared to rats treated with water prenatally).

**Figure 5 brainsci-07-00125-f005:**
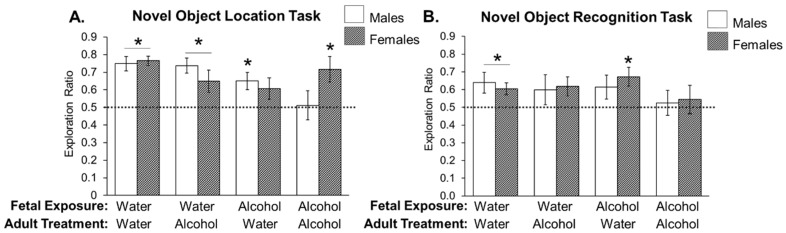
Impact of alcohol exposure (prenatal and adult) on hippocampal-dependent (Novel Object Location) and non-hippocampal-dependent (Novel Object Recognition) learning and memory in male and female rats. Pregnant dams were treated with alcohol or water on E10-16 (2 g/kg, 2×/day). Offspring were allowed to grow up undisturbed at which point they were taken through the Novel Object Location task and then one week later, the Novel Object Recognition task. The protocol for these tasks is depicted in Figure 1. (**A**) The average discrimination ratio of the time spent exploring the novel location during the three minute task is presented here. * *p* < 0.05 indicates discrimination ratios significantly greater than chance value (0.5), which is indicative of learning and memory in the task. In particular, all treatment groups were able to learn the Novel Object Location task except females exposed to alcohol prenatally and males exposed to alcohol prenatally and again in adulthood. (**B**) The average discrimination ratio of the time spent exploring the novel object during the three minute task is presented here. * *p* < 0.05 indicates discrimination ratios significantly greater than chance value (0.5), which is indicative of learning and memory in the task. In particular, only water-treated male and female rats, and the female rats treated with alcohol prenatally were able to learn and remember the Novel Object Recognition task.

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
