# Peer review of "Impact of Prenatal and Subsequent Adult Alcohol Exposure on Pro-Inflammatory Cytokine Expression in Brain Regions Necessary for Simple Recognition Memory"

_brainsci, 2017, doi:10.3390/brainsci7100125_

Round 1

Reviewer 1 Report

The manuscript by Terasaki and Schwarz investigates the cytokine expression in rats with prenatal and subsequent adult alcohol exposure. Pregnant rats are exposed to low alcohol treatment chronically. Male and female offspring is then given an acute alcohol exposure. The cytokine IL-1beta and IL-6 are found to be altered in the hippocampus, prefrontal cortex and perirhinal cortex. The authors also show that the expression levels of microglia marker CD11b and growth factor BDNF are changed in these brain regions. Behavioral tests further confirmed that the hippocampus-dependent memory and prefrontal cortex-dependent cognitive function are disrupted in these rats. Overall, the manuscript is interesting and the experimental design is solid. However, the manuscript could be improved based on the following comments.

The authors emphasize the role of microglia in the rats with prenatal and subsequent adult alcohol exposure. However, the evidence is indirect in this manuscript. Although the cytokines and the microglia marker are altered, the authors should provide direct and stronger data to show that microglia is changed, for example, using immunostaining to show that either the morphology or number of microglia is altered in different brain regions.  

The labeling of the figures are unclear. In fig 2, fig 3 and fig 4, star should be clearly labeled between groups instead of stating "relative to all other groups", which is not true. The labeling of Y axis is missing in B and C in fig 2-4.

Description of figure 2 should be modified. For example, the text describes that the CD11b is decreased in the rats. But the bar of the Alcohol/Alcohol female is obviously higher than the bar of the control. In addition, the BDNF level is not compared within male and female groups.

For the behavioral test (fig 5 and fig 6), the labeling and results are very confusing. Additionally, the authors fail to interpret the data and propose possible underlying mechanisms. Why the Alcohol/Water but not the Alcohol/Alcohol female shows deficits in the novel object location task? Is it related to the function of microglia? Why there is gender difference in these tasks?

The changes in cytokines, or microglia, or BDNF are not consistent with the changes in behaviors. In order to explain the behavioral phenotypes, the authors should provide evidence to support the idea that the alter microglia underlies the behavioral change, or perform rescue experiments.

Discussion should be modified. Instead of repeating the results, the authors should put more efforts on comparing their data to others, interpreting the results and discussing the contradictory points.     

Author Response

Brain Sciences #222930

Reviewer #1:

“The manuscript by Terasaki and Schwarz investigates the cytokine expression in rats with prenatal and subsequent adult alcohol exposure. Pregnant rats are exposed to low alcohol treatment chronically. Male and female offspring is then given an acute alcohol exposure. The cytokine IL-1beta and IL-6 are found to be altered in the hippocampus, prefrontal cortex and perirhinal cortex. The authors also show that the expression levels of microglia marker CD11b and growth factor BDNF are changed in these brain regions. Behavioral tests further confirmed that the hippocampus-dependent memory and prefrontal cortex-dependent cognitive function are disrupted in these rats. Overall, the manuscript is interesting and the experimental design is solid. However, the manuscript could be improved based on the following comments.”

“The authors emphasize the role of microglia in the rats with prenatal and subsequent adult alcohol exposure. However, the evidence is indirect in this manuscript. Although the cytokines and the microglia marker are altered, the authors should provide direct and stronger data to show that microglia are changed, for example, using immunostaining to show that either the morphology or number of microglia is altered in different brain regions.”  

Unfortunately, we are unable to complete the proposed immunohistochemistry experiments in the revision timeframe provided; however, we chose to analyze the expression of cytokines from the tissue that we collected because this approach provides us with a better understanding of the function of the immune cells in the brain, and how that may be altered by prenatal or adult alcohol exposure. Prior to collecting this tissue for the expression of cytokines, we perfuse the animal with cold 0.9% saline, which removes peripheral immune cells from the brain. Previous papers have shown that when neural tissue is perfused of peripheral immune cells, the resultant immune cell population is almost 96% microglia, with just a small percentage of perivascular macrophages (for example, Williamson et al., J. Neurosci, 2011).  Thus we are confident that the changes in gene expression that we are measuring are the result of changes in microglial function.  This important point for interpretation has been added to the Discussion (page 10, lines 405-410).  Moreover, it is often the case that changes in microglial function and increases in the expression of cytokines can occur even in the absence of any overt changes in cell number or morphology.  We have often seen this to be the case, and thus we feel confident that we took the best approach to measure possible changes in the function of microglia that may in turn impact neural function and behavior.

“The labeling of the figures are unclear. In fig 2, fig 3 and fig 4, star should be clearly labeled between groups instead of stating "relative to all other groups", which is not true. The labeling of Y axis is missing in B and C in fig 2-4.”

Thank you for the suggestions.  We have significantly modified the figure legends, including description of each symbol and its meaning. We hope that the changes improve the clarity of the figure legends.  We have also included the Y axes that were missing in panels B and C across Figures 2-4.

“Description of figure 2 should be modified. For example, the text describes that the CD11b is decreased in the rats. But the bar of the Alcohol/Alcohol female is obviously higher than the bar of the control. In addition, the BDNF level is not compared within male and female groups.”

The 2x2x2 ANOVA for CD11b expression in the hippocampus revealed only a main effect of adult alcohol treatment, such that treatment with alcohol significantly decreased CD11b mRNA. Given that our analysis revealed only this main effect, post hoc tests were not run across other groups for individual comparisons. The 2x2x2 ANOVA for BDNF expression in the hippocampus revealed a significant interaction of sex and prenatal alcohol exposure.  Post hoc comparisons revealed only that prenatal alcohol exposure increased BDNF expression in the female rats relative to the BDNF expression in the male rats exposed to prenatal alcohol (in the males it appears to go down slightly following prenatal alcohol exposure). Notably, these levels in females and males that had been exposed to alcohol prenatally appear to be on opposite ends of a “spectrum”; however, post hoc comparisons revealed that BDNF expression in the females exposed to alcohol prenatally were not significantly higher than either control, water-treated males or females, and similarly BDNF expression in the males exposed to alcohol prenatally were not significantly lower than either control, water-treated males or control females. There was also no sex difference in the expression of BDNF in prenatally water-treated males vs. females.

“For the behavioral test (fig 5 and fig 6), the labeling and results are very confusing. Additionally, the authors fail to interpret the data and propose possible underlying mechanisms. Why do the Alcohol/Water but not the Alcohol/Alcohol females show deficits in the novel object location task? Is it related to the function of microglia? Why there is gender difference in these tasks?”

After looking at the behavioral data in the preparation of this revised manuscript, we ultimately decided to completely revise the presentation of this data in order to make the data and the interpretation of these results (hopefully) less confusing.  We also combined the previous Figure 5 and 6 into one Figure 5 (A and B) for easier comparison across the two cognitive tasks. 

The second question raised, “Why do the Alcohol/Water but not the Alcohol/Alcohol females show deficits in the novel object location task? Is it related to the function of microglia?” is an excellent one, and we have addressed this in the revised Discussion (page 11, lines 439-455).

We don’t see overall sex differences in these cognitive tasks – specifically there were no significant main effects of sex in the NOR or NOL analyses; however, we have found a significant interaction of sex and alcohol exposure, either prenatally or in adulthood and we have addressed these interesting findings in the revised Discussion.

“The changes in cytokines, or microglia, or BDNF are not consistent with the changes in behaviors. In order to explain the behavioral phenotypes, the authors should provide evidence to support the idea that the alterations in microglia underlies the behavioral change, or perform rescue experiments. Discussion should be modified. Instead of repeating the results, the authors should put more efforts on comparing their data to others, interpreting the results and discussing the contradictory points.”   

The Discussion has been significantly revised in order to address many of these concerns and the concerns of Reviewer #2.  Unfortunately, the data are not always easy to interpret, though we feel they are quite informative. Upon further reflection we see some interesting effects in the data that we have expanded upon in the revised Discussion.  Unfortunately, there was not additional time to run a “rescue” experiment, though we think this would be a wonderful future experiment wherein minocycline could be administered to block the activation of microglia, either at the time of the prenatal alcohol exposure or later in life, at the time of the adult alcohol treatment. This would allow one to conclude that blocking microglia and the associated changes in cytokines can prevent the deficits in learning and memory that seem to be caused by the alcohol exposures.  We have added a Discussion of future experiments in the revised manuscript (page 11, lines 452-455).

Reviewer 2 Report

Summary

In this manuscript, the authors describe the effects of prenatal low ethanol exposure with or without adult binge exposure on the gene expression of IL-1β, IL-6, CD11b, and BDNF in several adult brain regions. The effects of the ethanol exposure on behavioral tests (novel object location and novel object recognition memory tests) were also examined. The aim of this study is important because the long-lasting effects of rather low levels of prenatal ethanol have not been well explored. While their data indicated some of the effects of prenatal low ethanol with or without adult binge exposure, the interpretation of the results should be more careful especially regarding the relationship between gene expression, inflammation, and memory-related behavioral functions. Specific comments are as follows:

Comments

Introduction, line 40: Reference 6 seems inappropriate here.

Materials and Methods, line 78: the period of ethanol administration, E10-E16, should be stated although there is a sentence, “this week-long treatment.” in line 82.

Materials and Methods, line 159: “Examination of NOR memory following acute alcohol exposure during a 24 hour delay” should be “2.7 Novel Object Recognition Task”.

Discussion, line 353: BDNF seems to be increased at least in perirhinal cortex.

Discussion, line 354: Although levels of cytokines change by ethanol treatment, it is difficult to suggest that “alcohol is acting as an inflammatory agent and causing a selective increase in levels of IL-6” because IL-1β and microglial marker CD11b decreased. Also, BDNF increased. Is there any other explanation?

Discussion, line 402: From the present study, it is difficult to correlate alcohol-induced neuroinflammation with deficits in memory tasks, probably because there are many other long-term effects of prenatal ethanol which affect adult behavior such as memory functions.

Fig. 2 legend, line 235: The legend says “*p<0.05 relative to all other groups, while **p<0.01 relative to all other groups”. This means that even gene expression of male IL-6 (water/alcohol) is significantly different from male IL-6 (alcohol/alcohol) in Fig. 1B, for example? The same question is applied to Fig. 3 legend, line 254 and Fig. 4 legend, line 273.

Fig. 5 and Fig. 6: The order of the legend (water/water, water/alcohol, alcohol/water,      alcohol/alcohol) should be the same as the order of the graph bars.

Author Response

Brain Sciences #222930

Reviewer #2

“In this manuscript, the authors describe the effects of prenatal low ethanol exposure with or without adult binge exposure on the gene expression of IL-1β, IL-6, CD11b, and BDNF in several adult brain regions. The effects of the ethanol exposure on behavioral tests (novel object location and novel object recognition memory tests) were also examined. The aim of this study is important because the long-lasting effects of rather low levels of prenatal ethanol have not been well explored. While their data indicated some of the effects of prenatal low ethanol with or without adult binge exposure, the interpretation of the results should be more careful especially regarding the relationship between gene expression, inflammation, and memory-related behavioral functions. Specific comments are as follows:

Thank you for the feedback and the comments below.  The Discussion has been edited to more carefully discuss the relationship between gene expression and memory-related functions, without overstating the interpretation. 

Comments

“Introduction, line 40: Reference 6 seems inappropriate here.”

Yes, this reference has been moved to an appropriate location in the revised manuscript, as it refers to the role of microglia in Alzheimer’s disease.

“Materials and Methods, line 78: the period of ethanol administration, E10-E16, should be stated although there is a sentence, “this week-long treatment.” in line 82.”

We apologize for the oversight.  We have indicated in the revised version that the treatment with prenatal alcohol occurred from E10-E16 (Page 2, Line 78).

“Materials and Methods, line 159: “Examination of NOR memory following acute alcohol exposure during a 24 hour delay” should be “2.7 Novel Object Recognition Task”.”

Thank you for noting this.  We have included 2.7 as a header on Page 4, Line 159.

“Discussion, line 353: BDNF seems to be increased at least in perirhinal cortex.”

Discussion, line 354: Although levels of cytokines change by ethanol treatment, it is difficult to suggest that “alcohol is acting as an inflammatory agent and causing a selective increase in levels of IL-6” because IL-1β and microglial marker CD11b decreased. Also, BDNF increased. Is there any other explanation?

Discussion, line 402: From the present study, it is difficult to correlate alcohol-induced neuroinflammation with deficits in memory tasks, probably because there are many other long-term effects of prenatal ethanol which affect adult behavior such as memory functions.

“Fig. 2 legend, line 235: The legend says “*p<0.05 relative to all other groups, while **p<0.01 relative to all other groups”. This means that even gene expression of male IL-6 (water/alcohol) is significantly different from male IL-6 (alcohol/alcohol) in Fig. 1B, for example? The same question is applied to Fig. 3 legend, line 254 and Fig. 4 legend, line 273.”

This was also a concern of Reviewer #1.  All figure legends have been edited to address these concerns.

“Fig. 5 and Fig. 6: The order of the legend (water/water, water/alcohol, alcohol/water, alcohol/alcohol) should be the same as the order of the graph bars.”

We apologize for the previous oversight.  We have fixed Figures 5 and 6 and in fact after looking at the data again, we thought the data would be easier to interpret if we changed the way in which the data are graphed.  We also combined the previous Figure 5 and 6 into one Figure 5 (A and B) for easier comparison across the two cognitive tasks.

Round 2

Reviewer 1 Report

The authors answered my comments.